# The Louisiana Amphibian Monitoring Program from 1997 to 2017: Results, analyses, and lessons learned

**Jacoby Carter**[1]*, **Darren Johnson**[2], **Jeff Boundy**[3], **William Vermillion**[4]

**1** U.S. Geological Survey, Wetland and Aquatic Research Center, Lafayette, Louisiana, United States of America, **2** Cherokee Nation Technologies under contract to the U.S. Geological Survey, Lafayette, Louisiana, United States of America, **3** Natural Heritage Program, Louisiana Department of Wildlife and Fisheries (Retired), Baton Rouge, Louisiana, United States of America, **4** Gulf Coast Joint Venture, US Fish and Wildlife Service, Lafayette, Louisiana, United States of America

* carterj@usgs.gov

**Data Availability Statement:** The data is currently available in the USGS public data base repository ScienceBase: https://doi.org/10.5066/P9VNBWM2. The doi is referenced in the paper with the following citation: Carter J., Louisiana Amphibian

## Abstract

To determine trends in either frog distribution or abundance in the State of Louisiana, we reviewed and analyzed frog call data from the Louisiana Amphibian Monitoring Program (LAMP). The data were collected between 1997 and 2017 using North American Amphibian Monitoring Program protocols. Louisiana was divided into three survey regions for administration and analysis: the Florida Parishes, and 2 areas west of the Florida parishes called North and South. Fifty-four routes were surveyed with over 12,792 stops and 1,066 hours of observation. Observers heard 26 species of the 31 species reported to be in Louisiana. Three of the species not heard were natives with ranges that did not overlap with survey routes. The other two species were introduced species, the Rio Grande Chirping Frog (*Eleutherodactylus cystignathoides)* and the Cuban Treefrog (*Osteopilus septentrionalis)*. Both seem to be limited to urban areas with little to no route coverage. The 15 most commonly occurring species were examined in detail using the percentage of stops at which they observed along a given survey and their call indices. Most species exhibited a multimodal, concave, or convex pattern of abundance over a 15-year period. Among LAMP survey regions, none of the species had synchronous population trends. Only one group of species, winter callers, regularly co-occur. Based on the species lists, the North region could be seen as a subset of the South. However, based on relative abundance, the North was more similar to Florida parishes for both the winter and summer survey runs. Our analyses demonstrate that long-term monitoring (10 years or more) may be necessary to determine population and occupancy trends, and that frog species may have different local demographic patterns across large geographic areas.

## Introduction

In the 1990's, concerns over perceived declines in North American amphibian populations led the members of the Declining Amphibian Population Task Force to recommend the

Monitoring Program Survey Frog Call Observation Data: 1997-2017. U.S. Geological Survey data release https://doi.org/10.5066/P9VNBWM2.

**Funding:** The authors received no specific funding for this work.

**Competing interests:** The authors have declared that no competing interests exist.

implementation of a statistically defensible amphibian monitoring program [1]. In 1997 the North American Amphibian Monitoring Program (NAAMP) was established to answer the question, 'Are these perceived declines real?" To address this question, NAAMP utilized a network of largely volunteer observers who monitored calling frogs along roadside routes, similar to the Breeding Bird Survey [2]. From 1997 to 2015 the U.S. Geological Survey (USGS) coordinated NAAMP. The USGS developed the monitoring protocol, provided the states with randomized route starting points, and stored data provided by the states. The states recruited and trained agency personnel and volunteers to conduct the surveys. In 2015, the USGS terminated the NAAMP.

The Louisiana Amphibian Monitoring Program (LAMP) is Louisiana's state 'chapter' of NAAMP. The purpose of the LAMP is to determine if there are changes in frog distribution and abundance in the state of Louisiana over time. The first LAMP surveys were conducted in 1997, and the program is still active as of 2021. Fifty-nine routes were set up across the state, 54 of which were surveyed one or more times from 1997 through 2017. Using data collected with NAAMP protocols we want to determine if there were changes in frog distribution and abundance in Louisiana over the observation period. We had three broad questions: (1) 'Can we use call data to detect trends in frog populations?' (2) 'Are there trends in frog populations?' and (3) 'If there are trends, are these associated with frog communities as a whole or with individual species, or both?'

We wanted to look at the above question at two hierarchical levels: for frog communities as a whole and for individual species. Finally, we wanted to use LAMP data to describe Louisiana's anuran communities. Our community- and species-level questions were as follows:

Community Level Questions:

1. Are there significant changes in species richness over time for a given route or in a given region?

2. Are there significant changes in species call index (as a proxy for abundance) over time for a given route or in a given region? That is, independent of species richness, are more or fewer frogs calling?

3. Do some species co-occur and can they be considered as a community or as indicators of a particular habitat?

4. Do the different LAMP regions have different community compositions?

Species Level Questions:

1. Which species were detected during the surveys, where, and when?

2. Which species were the most observed and which have the highest abundance (call index) when observed?

3. Did any species show a change in frequency of observations or abundances (call index) for a given region, route, or time of year?

## Methods

LAMP data used in this study can be downloaded from the USGS's ScienceBase data server [3].

### Survey routes

Thirty-seven (37) of the 64 parishes (i.e., 'counties') in Louisiana have one or more survey routes. Starting with locations and directions randomly selected by the NAAMP program,

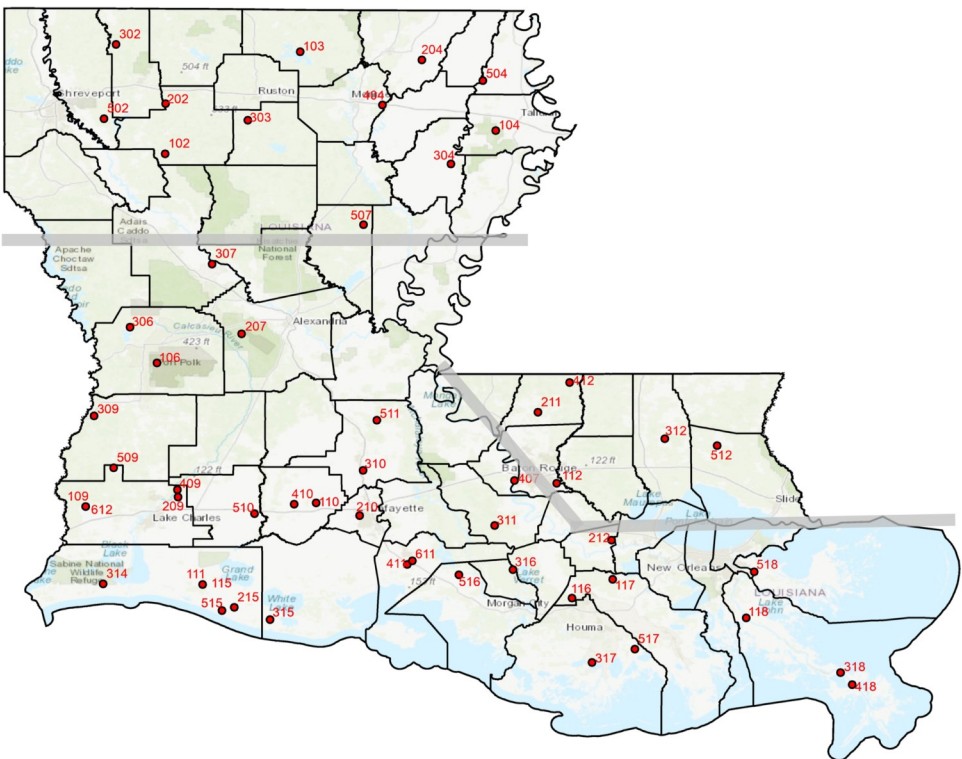

**Fig 1. Approximate locations of LAMP routes and regions in this study.** The dots show approximate route locations. The numbers are the last three digits of the 6-digit NAAMP route number. The broad gray lines delineate the regions and the black lines are parish (i.e., county) or state boundaries. The South and Florida regions used the same observation windows (see Table 2). The Florida region is comprised of the 'Florida Parishes', locations east or north of the Mississippi River. The areas west of the Mississippi River are divided at approximately 31˚41'N latitude. See S1 Table for a list of full NAAMP route numbers and the latitude and longitude of the first stop.

LAMP volunteers set up survey routes with 10 stops each. Once established, survey routes and stops can only be changed for reasons of safety or compliance with protocols. S1 Table lists LAMP Route names, NAAMP numbers, and the latitude and longitude of first stop. Fig 1 is a state map showing LAMP route starting points.

The first surveys were conducted in 1997. The total number of stops observed varied between routes for two reasons. Not all routes were surveyed every year, or completely surveyed in a given year. A subset of 54 of the 59 established routes were deemed to have sufficient data for analysis (Table 1). The state was divided into three regions for administrative and analysis purposes based on what was known about frog and toad distributions and regional phenology at the time the survey was initiated. The regions are the Florida Parishes, North and South (Fig 1). The Florida Parishes region (henceforth, 'Florida region') is north and east of the Mississippi River and north of Lake Pontchartrain and Lake Maurepas. The area of the state west of the Mississippi River is divided between North and South at latitude 31˚41' N. This line reflects a geographical break in the randomly assigned route locations between north and south Louisiana, and the 16˚C average annual temperature line [4], where north of that line the winters are colder, longer, and frogs start calling later in the year. The 40-day survey windows (or 'runs') for the South and Florida regions are the same; the survey windows for the North begin about 30 days later than those in the South due to their colder climate (Table 2).

The LAMP followed the NAAMP survey protocols [5]. Each survey was initiated at least ½ hour after sunset and before midnight. The surveys were run three times a year during

**Table 1. LAMP routes.**

| Route Name | Number of Stops | Region | Analysis | Route Name | Number of Stops | Region | Analysis |
|---|---|---|---|---|---|---|---|
| Ossun | 598 | South | S, C, T, G | Loranger | 387 | Florida | S, C, T, G |
| Prairie Laurent | 590 | South | S, C, T, G | Woodland | 386 | Florida | S, C, T, G |
| Charenton | 550 | South | S, C, T, G | McManus | 360 | Florida | S, C, T, G |
| Rayne | 540 | South | S, C, T, G | Tiger Bend | 177 | Florida | S, C |
| Egan | 474 | South | S, C, T, G | Blond | 170 | Florida | S, C |
| Brannon | 418 | South | S, C, T, G | Cotton Valley | 570 | North | S, C, T, G |
| Bayou Jack | 410 | South | S, C, T, G | Koran | 530 | North | S, C, T, G |
| Pickering | 410 | South | S, C, T, G | Roy | 450 | North | S, C, T |
| Anacoco | 390 | South | S, C, T, G | Ada | 270 | North | S, C |
| Bayou Sorrel | 358 | South | S, C, T, G | Rocky Branch | 210 | North | S, C, T, G |
| Gramercy | 340 | South | S, C, T, G | Horseshoe Lake | 140 | North | S, C |
| Otis | 270 | South | S, C, T, G | Tensas | 100 | North | S, C |
| Falgout Canal | 250 | South | S, C, T | Monticello | 80 | North | S, C |
| Palmetto | 245 | South | S, C | Bayou Funny Louis | 70 | North | S, C |
| Belle River | 240 | South | S, C | Mill Haven | 70 | North | S, C |
| Hecker | 240 | South | S, C | Boggy Womble | 30 | North | S, C |
| Price Lake | 220 | South | S, C | Ansley | 20 | North | S |
| Lake Fourteen | 210 | South | S, C | | | | |
| Holly Beach | 207 | South | S, C | | | | |
| Le Bleu | 190 | South | S, C | | | | |
| Headquarter Canal | 182 | South | S, C | | | | |
| De Quincy | 180 | South | S, C | | | | |
| Jennings | 180 | South | S, C, T | | | | |
| Montegut | 150 | South | S, C | | | | |
| Merryville | 140 | South | S, C | | | | |
| Big Woods | 130 | South | S, C | | | | |
| Antonia | 120 | South | S, C | | | | |
| Choctaw | 80 | South | S, C | | | | |
| Old Brannon | 80 | South | S | | | | |
| Little Chenier | 70 | South | S, C | | | | |
| Venice | 70 | South | S | | | | |
| Boothville | 60 | South | S | | | | |
| Phoenix | 50 | South | S | | | | |
| Violet | 40 | South | S | | | | |
| Wine Bayou | 40 | South | S | | | | |
| Odra | 30 | South | S | | | | |
| Old Blonde | 20 | South | S | | | | |

Route name, number of stops, LAMP region, and what analysis the data from that route were used for. The list is sorted by region and number of stops. See Fig 1. for a map of the approximate locations of the routes. See S1 Table for full NAAMP route number, and the first stop's latitude and longitude. Key S- species richness, season, and call intensity analysis; C- species co-occurrence analysis; T- linear trend analysis, G- GAM analysis.

The base map is from ESRI online (for a description of the copyright information see: http://goto.arcgisonline.com/maps/World_Topo_Map), and is in turn based on USGS topo data, and is not copyrighted. The base map is a state topo map with parish (i.e., county) boundaries and larger geographic features labeled. ArcInfo was used to superimpose on the public domain base map the locations of NAAMP route survey routes and then to generate a TIFF file of the new map.

The LAMP regional boundary lines were overlaid on the TIFF file using MS PowerPoint software and then saved as a new TIFF file. The coordinates used are listed in S1 Table.

**Table 2. Louisiana Amphibian Monitoring Program (LAMP) survey windows and minimum temperatures.**

| Region | First Run (Winter) | Second Run (Spring) | Third Run (Summer) |
|---|---|---|---|
| North | January 27-March 8 | March 27-May 7 | May 7 –July 7 |
| South and Florida | January 1 –February 10 | February 26-April 7 | April 27-June 5 |
| Minimum Temperature[&] | 5.6˚ C | 10˚C | 12.8˚C |

Note: The air temperature must be at or above the minimum temperature at the start of a survey.

windows meant to capture 'winter', 'spring' or 'summer' calling frog species. The observers did not need permits or approvals to conduct their surveys. All routes and stops were on public roads along right-of-ways, and didn't require landowner permissions. No permits were required by either state or federal agencies and no endangered or protected species were interacted with. Volunteers only passively listened for frogs and did not come into contact with them nor call to them in order to get a response. These observation windows were called Runs '1', '2', and '3' respectively. Each stop was considered a separate observation and ideally there were a total of 30 observations per route, per year. The frog call observers listened for 5-minutes at each stop, logged frog species heard calling, and assessed an index of calling activity for each species. Using the stratified by habitat protocol [5] stops were spaced a minimum of 0.8 km apart so that the same individuals were not heard at neighboring stops. As a quality control measure, observers were required to take an online test on their ability to identify frog calls and rank calling activity [6]. In addition to the frog calling data, observers reported the time of day the survey was conducted and recent weather conditions. Data sheets from survey runs were forwarded to the Louisiana state coordinator (J. Boundy) who performed quality control and entered the data into the national database.

## Route data selection

The routes used depended on the analysis (Table 1). All stops with one or more species calling were used for analyses of species richness, phenology, and average call intensity (54 routes met these conditions). All stops with two or more species calling were used for co-occurrence analyses (45 routes met these conditions). For trend analysis using general linear models, routes had to have data sets from 8 or more different years, and the surveys needed to span 20 or more years (21 routes met these conditions). For General Additive Model (GAM) analysis route data had to have data sets from 8 or more different years, surveys spanning 20 or more years, and no gaps in the data of more than 5 years (19 routes met these conditions).

The data were examined at different hierarchical levels in both space and time. In all cases time (years) was the independent variable. When all three runs per year were combined data are referred to as "route" or "region". "Region" data included data from all routes in a given region. When data were examined by using all the routes in a given region for a given run, data are referred to as "region-run" data (3 regions, 3 runs per year, 9 possible combinations). When data were examined looking at individual routes separated by run, it is referred to as "route-run" data (54 routes, 3 runs, 162 possible combinations). Individual species trends were examined by both region-run and route-run (171 possible combinations).

## Species selected for analysis

All species and all observations were used to determine when species called and where they were observed. A species required a minimum of 200 observations for trend analyses. For a

species route-run trend analysis the species needed to be detected over 8 or more years on the route-run.

## Route and region: Species richness and average call index

Species richness for each stop was the number of species heard. Species richness for a route-run was the number of different species observed over all stops during a run. The average species richness for a run-route was the average species richness of all stops along the route, including stops where no frogs were observed. We used a linear model to estimate the change (slope) in route species richness over the 1997–2017 observation period for all the routes by run for each region.

The average call index (ACI) for a route-run was the average of all call indices of all species observed. Stops where no frogs were heard calling were not used in calculating the average. Species richness and ACIs were calculated for both the regions and routes, and by run. We used a general linear model to estimate the change (slope) in ACI over the 1997–2017 observation period.

## Regional assessment of species co-occurrence

A Two-Way Species Indicator Analysis (TWINSPAN) in PC Ord software [7] was used to determine if groups of species formed 'communities' of co-associations [8,9]. In this analysis each stop was treated as a separate observation and only observations with two or more species were used. The species used for this analysis also needed a minimum of 10 observations from routes with 5 or more years of observation for that species.

## Relative abundance analysis

A Pielou's Evenness Index (J) is a measure that relates species richness (i.e., the number of other categories) to their relative abundances [10]. J is the ratio of the measured Shannon diversity index (Eq 1) to perfectly even abundances (Eq 2). If J equals 1, all species are present in equal abundance. The smaller J, the more skewed the observations.

$$H = \sum p_i * \ln(p_i), \qquad \text{Eq1}$$

for i = 1 to R; where $p_i$ is the relative abundance of species or
    categories 'i', and
    R is the total number of species or categories

$$J = H/H_{max}; \qquad \text{Eq2}$$

where $H_{max} = \ln(R)$

J was calculated for the species observations statewide and by region-run. J was also used to compare the evenness of the sampling effort between regions using the routes as categories and number of stops as observations.

## Species: Percent observation and average call index

The data were analyzed to determine if there were trends in: how often species were observed; species richness, or the species call abundance index. All runs in which a given species was observed were used to calculate that species' percent observation for that run. For example, if a species was observed on three different stops percent observation for that run was 0.3. This analysis was done for regions by run, and for routes by run.

A species average call index (sACI) was the average of all non-zero call indices of that species on a route-run or region-run. A routes-run ACI was the average for all species calling on a given region and run. A region-run ACI was the average for all species calling on a given route and run. We used PROC REG in SAS 9.4 [11] for linear model analysis to look for trends in route-run and region-run ACIs. We use PROC GAM in SAS to look for species sACI trends by route-run and region-run.

## Generalized Additive Model (GAM) analysis

When we ran simple linear regressions for route-run and region-run species richness and ACI, the assumptions of normality and homogeneity were violated. Since those could not be fixed with a transformation, we ran a generalized additive model (GAM) using the generalized cross validation approach, whereby the estimated parameters are chosen via a generalized cross validation [12]. GAMs are used when there is an expectation that behavior along the dependent variable may not be linear or may change over time. The GAM procedure fits generalized additive models as defined by Hastie and Tibshirani [13]. PROC GAM in SAS 9.4 was used using non-parametric regression and smoothing [11].

We ran GAM analysis for the species with 200 or more observations (15). An individual route-run used in this analysis had to have at least 8 stops visited. A species needed a minimum of 8 or more route-run observations for GAM analysis of percent observations. Route-run combinations found to have significant trends then had their GAMs classified as one of the following trend patterns: increasing, decreasing, concave, convex, or multimodal.

## Results

### Route statistics

The number of routes surveyed, and the number of runs completed varied from year to year. Survey data for a given route might be incomplete with missed stops, runs, or even consecutive years missing. Additionally, even when a route-run combination was completed, there might be one or more stops where no frogs were heard. Finally, there was with a significant decline in the number of routes surveyed starting in 2016 (Fig 2).

Between 1997 and 2017, 75 observers surveyed 54 routes and 12,792 stops. Sampling was uneven between the regions (Table 1) with the South having the largest number of routes (37) and stops observed (8,772), followed by North (12 Routes/ 2,540 stops) and Florida (5 routes/ 1,480 stops). Between 1999 and 2015 the number of stops varied between 500 and 700 per year. Species richness at a given stop varied between 0 (i.e., no species heard at the stop) to 11 (Fig 3).

If we treat each route as a category (i.e., as a 'species') and the number of stops observed as abundances, we can use J to compare the evenness of sampling between the different LAMP regions. We can then ask the question "Were some regions more reliant on heavily sampled routes for their results than other regions?" The J for the three regions using the number of stops visited per route was 0.961, 0.844 and 0.930 for Florida, North, and South respectively. Thus, while the number of routes surveyed was different in each region; the sampling effort for the routes within a region was comparable between regions.

### Species statistics

Statewide: Twenty-six different species were reported (Table 3); the number of observations ranged from 2,868 for Southern Leopard Frogs (*Lithobates sphenocephalus*) to 4 for Eastern Spadefoot Toads (*Scaphiopus holbrookii*), and the sACI for a given species ranged between 1.1

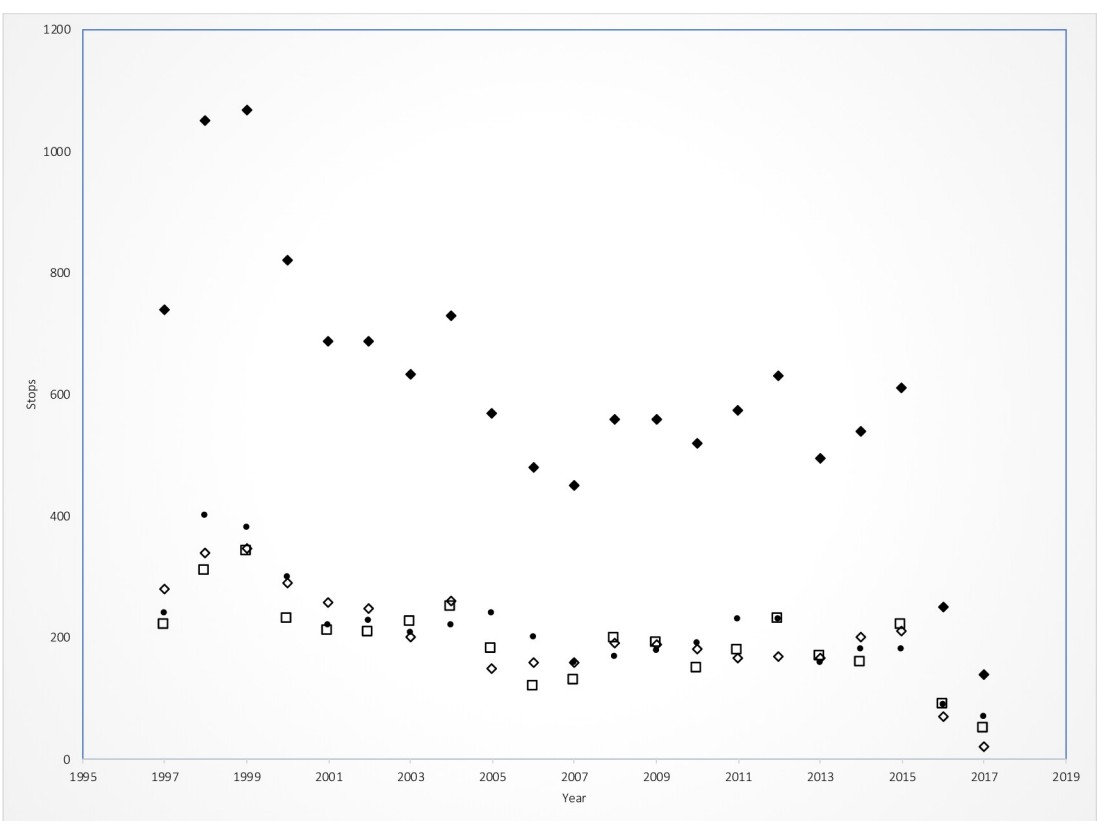

**Fig 2. Number of LAMP stops visited, 1997 through 2017 by year and run.** Key: solid-diamonds: sum of stops for year; solid-circles: Run 1 (winter); open-squares: Run 2 (Spring); open-diamonds: Run 3 (Summer).

and 3.0. Statewide, the frequency a species was observed was not a good predictor of average calling intensity (Fig 4). The taxonomy used here follows the Integrated Taxonomic Information System [14].

The frequency a given species was observed when plotted against its rank abundance was best modeled as an exponential function (Fig 5). J for the state overall was 0.30.

Species could be grouped by when they were heard calling as either winter callers, spring callers, or summer callers (Table 4). Cajun Chorus Frogs (*Pseudacris fouquettei*), Spring Peepers (*P. crucifer*), *L. sphenocephalus*, Crawfish Frogs (*L. areolatus*) and Pickerel Frogs (*L palustris*) were winter (e.g., Run 1) callers. Two toad species were primarily spring (e.g., Run 2) callers: American Toads (*Anaxyrus americanus*) and Oak Toads (*Anaxyrus quercicus*). Southern (*Acris gryllus*) and Northern Cricket Frogs (*Acris crepitans)* were almost evenly divided between spring and summer, and the rest of the species were observed calling 62% or more of the time during the summer (e.g., Run 3). One summer species, *S. holbrookii*, was never heard during the spring Run.

All species that called in winter also called in spring or summer, while 2/3 of species classified as Spring or Summer callers did not call in winter. Because of the above observations in addition to statewide trend analyses, some trends were analyzed by region, or region and run. Since there were only two spring calling species, we combined the spring and summer runs for some analyses (see below).

There were 5 species reported to be in Louisiana that were not observed: 3 native species, the Dusky Gopher Frog (*L. sevosa*) Ornate Chorus Frog (*P. ornata*) and Strecker's Chorus

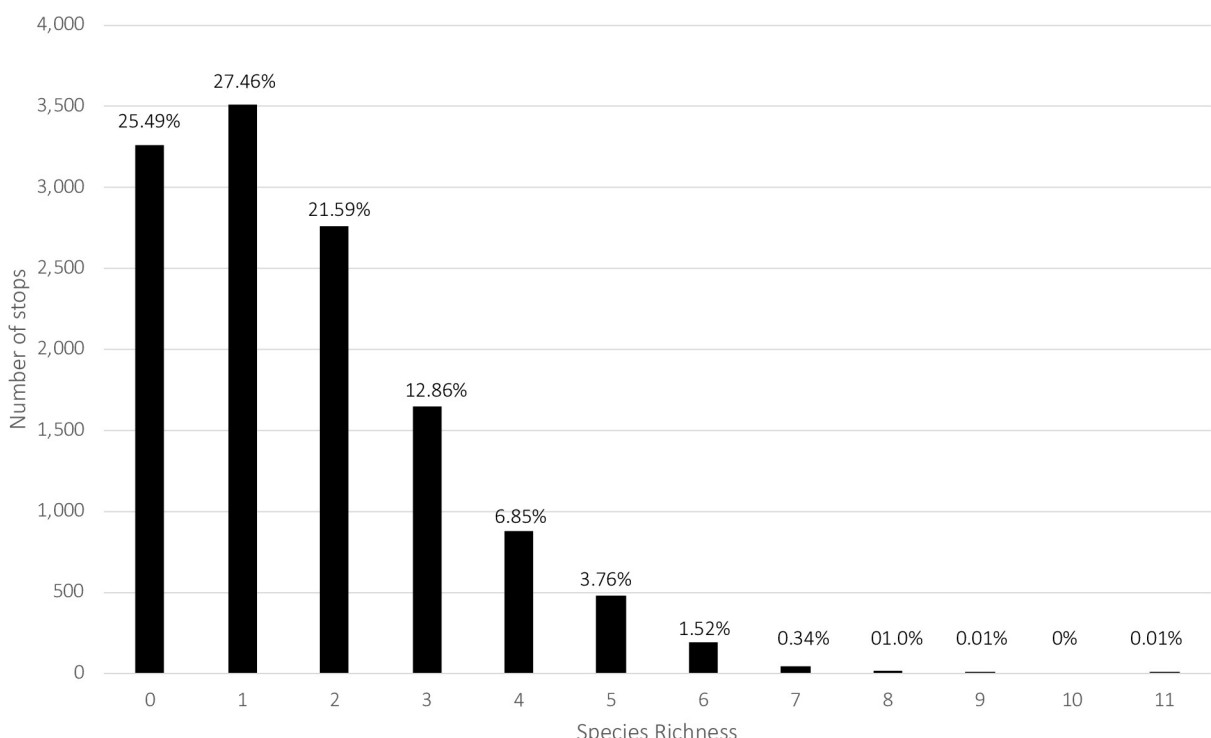

**Fig 3. Species richness frequency distribution of all survey stops.** The x-axis is the species richness, the y-axis is the number of stops with that given species richness. The percentage is the percentage of all stops with a given species richness.

Frog (*P. streckeri*); and 2 exotic species, the Rio Grande Chirping Frog (*Eleutherodactylus cystignathoides*), and the Cuban Treefrog (*Osteopilus septentrionalis*. Distribution maps [15,16] indicate that the documented locations of the *L. sevosa*, *P. ornata* and *P. streckeri* were not in the vicinity of LAMP routes. Also, *L. sevosa* and *P. ornata* are considered extirpated from the state, not having been seen since the 1960's [16]. *E. cystignathoides* has been documented in urban areas around Baton Rouge, Lafayette, Lake Charles, Alexandria, and Shreveport [17] since the early 1990's (J. Boundy, pers. obs.). *O. septentrionalis* has been documented in the cities of Baton Rouge, Lafayette and New Orleans since 2018 [18]. Neither species was observed on any survey route as of 2017.

Region: Florida, South, and North had 23, 22, and 17 species in their surveys respectively (Table 5). The most frequently observed species in the South was *L. sphenocephalus* and for Florida and North, *P. crucifer*. Both are winter calling species.

When we examined abundance by region and run (Table 6) the most commonly observed species for the regions as a whole were winter callers. For the spring and summer runs, the most common species were Green Treefrogs (*Dryophytes cinereus*) in the Florida region, Cope's Gray Treefrog (*D. chrysoscelis*) in the North region, and *A. crepitans* in the South region. The Pielou evenness index, J, for species observations was more even for Florida (0.8255) than for North (0.7935) and South (0.7940). Winter runs were consistently more uneven, meaning their observations were dominated by fewer species.

### Frequency observed and call intensity

**Region analysis.** The ACI varied year to year (Table 7). The only region-run combination with an R-squared greater than 0.3 was Florida-Run 2. It appeared to have a sustained decrease in average call intensity after 2004 (Fig 6).

**Table 3. List of species observed.**

| Rank | Common Name | Scientific Name | Observations | sACI |
|---|---|---|---|---|
| 1 | Southern Leopard Frog | *Lithobates sphenocephalus* | 2868 | 1.8 |
| 2 | Green Treefrog | *Dryophytes cinereus* | 2827 | 2.6 |
| 3 | Spring Peeper | *Pseudacris crucifer* | 2592 | 2.2 |
| 4 | Northern Cricket Frog | *Acris crepitans* | 2455 | 2.4 |
| 5 | Bronze Frog | *Lithobates clamitans* | 1823 | 1.3 |
| 6 | Cajun Chorus Frog | *Pseudacris fouquettei* | 1718 | 1.9 |
| 7 | Cope's Gray Treefrog | *Dryophytes chrysoscelis* | 1654 | 1.9 |
| 8 | Gulf Coast Toad | *Incilius nebulifer* | 1372 | 2.0 |
| 9 | American Bullfrog | *Lithobates catesbeianus* | 1189 | 1.1 |
| 10 | Squirrel Treefrog | *Dryophytes squirellus* | 842 | 1.8 |
| 11 | Fowler's Toad | *Anaxyrus fowleri* | 694 | 1.6 |
| 12 | Eastern Narrow-mouthed Toad | *Gastrophryne carolinensis* | 361 | 1.4 |
| 13 | Bird-voiced Treefrog | *Dryophytes avivoca* | 277 | 1.8 |
| 14 | Pig Frog | *Lithobates grylio* | 259 | 1.5 |
| 15 | Southern Cricket Frog | *Acris gryllus* | 219 | 2.1 |
| 16 | Gray Treefrog | *Dryophytes versicolor* | 104 | 2.7 |
| 17 | American Toad | *Anaxyrus americanus* | 65 | 1.7 |
| 18 | Greenhouse Frog | *Eleutherodactylus planirostris* | 56 | 1.2 |
| 19 | Southern Toad | *Anaxyrus terrestris* | 45 | 1.7 |
| 20 | Barking Treefrog | *Dryophytes gratiosus* | 25 | 2.2 |
| 21 | Pine Woods Treefrog | *Dryophytes femoralis* | 21 | 1.9 |
| 22 | Pickerel Frog | *Lithobates palustris* | 19 | 2.1 |
| 23 | Crawfish Frog | *Lithobates areolatus* | 11 | 1.3 |
| 24 | Hurter's Spadefoot | *Scaphiopus hurteri* | 7 | 1.3 |
| 24 | Oak Toad | *Anaxyrus quercicus* | 7 | 1.3 |
| 26 | Eastern Spadefoot | *Scaphiopus holbrookii* | 4 | 3.0 |

The list is sorted from most observed (Rank 1) to the fewest (26) statewide. The species average call index (sACI) is rounded to the tenth decimal place.

**Route analysis.** We regressed the ACI by year for the various route-run combinations. Of the 145 route-run combinations examined, 17 regressions with 10 or more years of observations had p-values of 0.05 or less, 10 had negative slopes and 7 positive slopes (Table 8). The R-squared values ranged from 0.2357 to 0.8445.

**Species richness.** *Regional trends.* North-Run 1 and the South-Run 3 had significant positive changes in mean species richness (Table 9). No other region-run combination had significant changes.

*Route-run trends.* Twenty-one routes had sufficient data to examine species richness. Using a p-value of 0.05 or less we found 16 route-runs with significance (Table 10). Five (5) route-run combinations had significant decreases in species richness and 11 had increases. Interestingly, in one case (Rayne), there was a decrease in Run 2 but an increase in Run 3.

## Two Way Indicator Species Analysis (TWINSPAN)

TWINSPAN was conducted separately for each LAMP region (Fig 7). The first group to separate out for all three regions were winter calling species (Run 1). Beyond the winter calling species no groupings were consistent between regions.

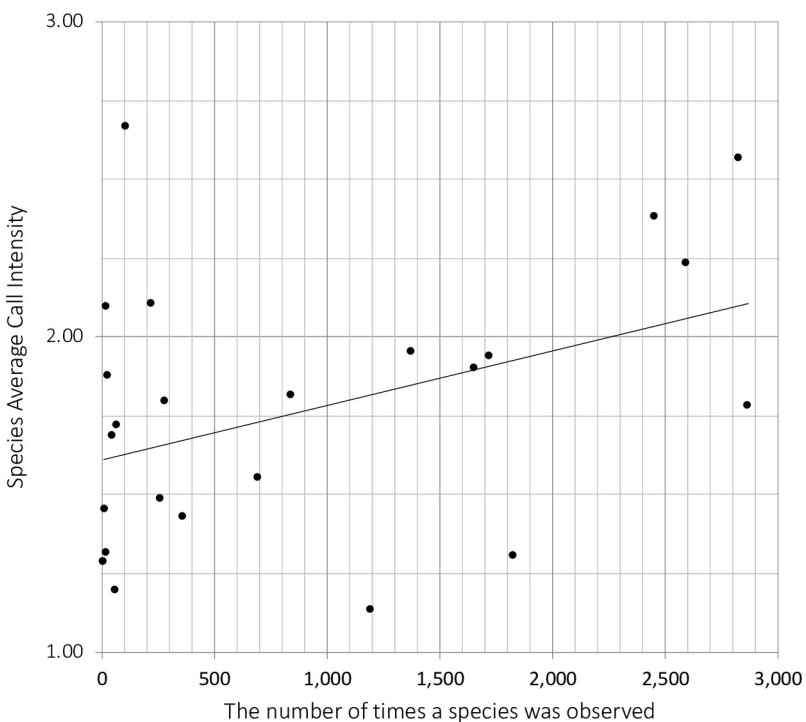

**Fig 4. The average calling intensity of a species versus frequency of its observation.** The regression of calling intensity on frequency was not significant. The equation of the line is: Y = 0.0002 X + 1.6106; $R^2$ = 0.1685.

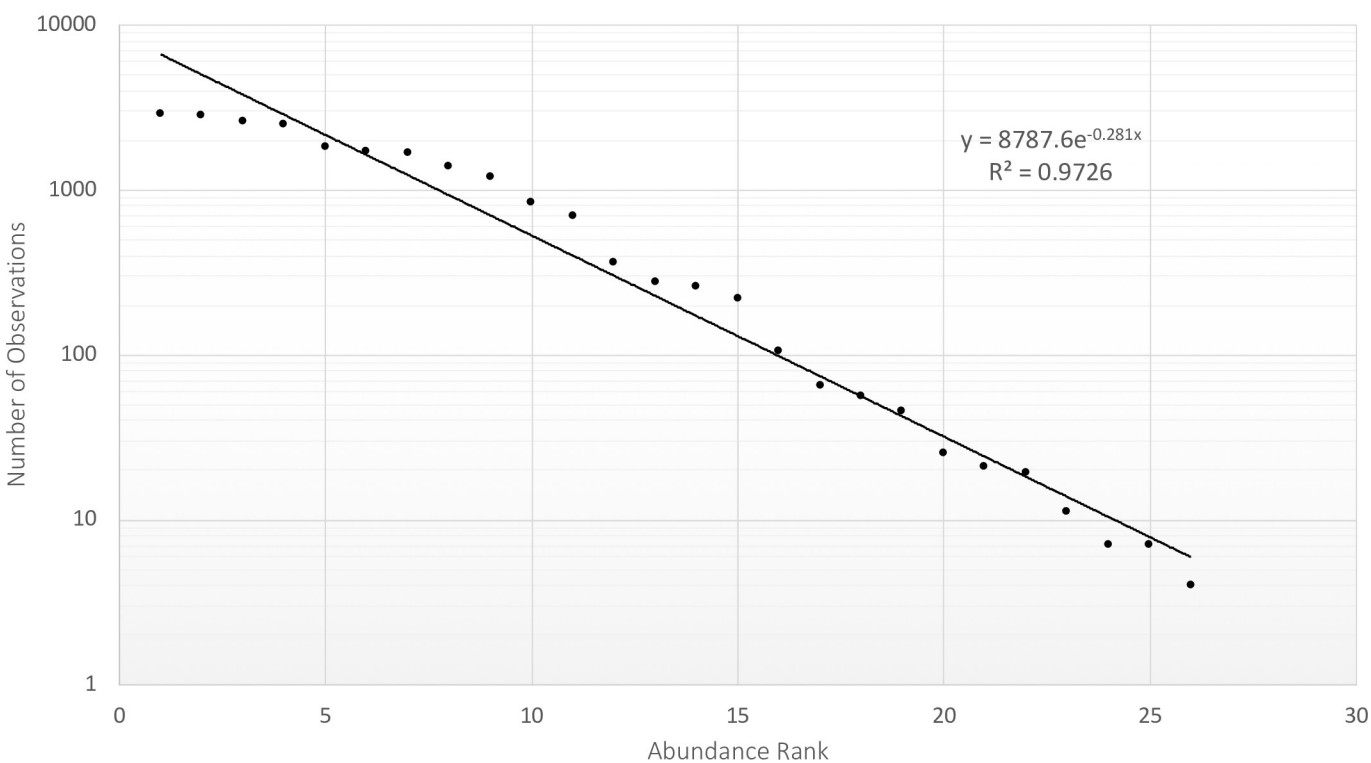

$$y = 8787.6e^{-0.281x}$$
$$R^2 = 0.9726$$

**Fig 5. Ranking of species from most observed (n = 2,868) to least observed (n = 4) on a logarithmic scale.** An exponential model best fits the data (see inset).

**Table 4. Percentage observation by run.**

| Species | % Run 1 | % Run 2 | % Run 3 | Dominate Season (Run) |
|---|---|---|---|---|
| *L. areolatus* | 91% | | 9% | Winter (Run 1) |
| *L. palustris* | 79% | 16% | 5% | Winter (Run 1) |
| *P. crucifer* | 74% | 25% | 1% | Winter (Run 1) |
| *P. fouquettei* | 70% | 26% | 4% | Winter (Run 1) |
| *L. sphenocephalus* | 55% | 34% | 11% | Winter (Run 1) |
| *A. americanus* | 2% | 74% | 25% | Spring (Run 2) |
| *A. quercicus* | | 71% | 29% | Spring (Run 2) |
| *A. crepitans* | 3% | 44% | 53% | Summer (Run 3) |
| *A. gryllus* | | 46% | 54% | Summer (Run 3) |
| *L. grylio* | 2% | 36% | 62% | Summer (Run 3) |
| *A. fowleri* | | 37% | 62% | Summer (Run 3) |
| *L. clamitans* | 1% | 36% | 63% | Summer (Run 3) |
| *L. catesbeianus* | | 36% | 64% | Summer (Run 3) |
| *D. chrysoscelis* | | 36% | 64% | Summer (Run 3) |
| *D. femoralis* | 10% | 24% | 67% | Summer (Run 3) |
| *D. avivoca* | | 29% | 70% | Summer (Run 3) |
| *E. planirostris* | 2% | 23% | 75% | Summer (Run 3) |
| *A. terrestris* | | 22% | 78% | Summer (Run 3) |
| *D. cinereus* | | 20% | 80% | Summer (Run 3) |
| *D. versicolor* | | 16% | 84% | Summer (Run 3) |
| *D. squirellus* | | 15% | 85% | Summer (Run 3) |
| *S. hurteri* | | 14% | 86% | Summer (Run 3) |
| *I. nebulifer* | 2% | 12% | 86% | Summer (Run 3) |
| *D. gratiosus* | | 8% | 92% | Summer (Run 3) |
| *G. carolinensis* | | 6% | 94% | Summer (Run 3) |
| *S. holbrookii* | | | 100% | Summer (Run 3) |

Values rounded to the nearest 1%. A species was placed into a season if 51% or more of its observations were in that season.

### Generalized Additive Model (GAM) analysis

We examined the 15 most common species along their possible route-run combinations over time to determine if they had changes in percent observation or their call index (as a proxy for abundance). Plots of the 12 species with 10 or more years' worth of data and significant GAMs ($p \leq 0.05$) were examined for model type (S1 File). There were 39 significant GAM models classified: multimodal = 32, convex = 6, concave = 9, and linearly increasing = 2 (Table 11). Seven (7) models showed an overall increase, and 1 an overall decrease over the observation period.

There were 9 regional models, and the remaining 30 models were route specific. At 12 each, North and Florida had relatively more models than South region's 15 models. This may be due to the fact that many South routes had less than 10 years' worth of survey data or breaks of 5 or more years between surveys, whereas the routes in the other two regions tended to have more complete data sets. At 10, the Bronze Frog (*L. clamitans*) had the greatest number of significant GAM models.

## Discussion

### Community level questions

As noted above, because different species tended to call during different times of year, the seasonal survey runs were examined independently.

**Table 5. Species abundance by LAMP region.**

| FLORIDA | | SOUTH | | NORTH | |
|---|---|---|---|---|---|
| **SPECIES** | **OBS** | **SPECIES** | **OBS** | **SPECIES** | **OBS** |
| *P. crucifer*[w] | 570* | *L. sphenocephalus*[w] | 2313* | *P. crucifer*[w] | 633* |
| *P. fouquettei*[w] | 308* | *A. crepitans* | 2176* | *D. chrysoscelis* | 616* |
| *D. cinereus* | 222* | *D. cinereus* | 2059* | *D. cinereus* | 546* |
| *A. gryllus* | 212* | *P. crucifer*[w] | 1389* | *P. fouquettei*[w] | 516* |
| *D. chrysoscelis* | 205* | *I. nebulifer* | 1219* | *L. clamitans* | 452 |
| *L. clamitans* | 196* | *L. clamitans* | 1175 | *L. sphenocephalus*[w] | 363 |
| *L. sphenocephalus*[w] | 192 | *L. catesbeianus* | 1004 | *A. fowleri* | 260 |
| *I. nebulifer* | 153 | *P. fouquettei*[w] | 894 | *A. crepitans* | 213 |
| *D. avivoca* | 111 | *D. chrysoscelis* | 833 | *L. catesbeianus* | 90 |
| *L. catesbeianus* | 95 | *D. squirellus* | 770 | *D. avivoca* | 76 |
| *A. crepitans* | 66 | *A. fowleri* | 396 | *G. carolinensis* | 52 |
| *A. americanus* | 61 | *G. carolinensis* | 277 | *D. versicolor* | 24 |
| *D. squirellus* | 55 | *L. grylio* | 251 | *D. squirellus* | 17 |
| *A. terrestris* | 41 | *D. avivoca* | 90 | *L. palustris*[w] | 10 |
| *A. fowleri* | 38 | *D. versicolor* | 73 | *A. americanus* | 3 |
| *G. carolinensis* | 32 | *E. planirostris*‡ | 55 | *S. hurteri*‡ | 1 |
| *D. gratiosus*† | 25 | *L. areolatus*[w]‡ | 11 | *E. planirostris*‡ | 1 |
| *D. femoralis*† | 21 | *A. gryllus* | 7 | | |
| *L. grylio* | 8 | *S. hurteri*‡ | 6 | | |
| *A. quercicus*† | 7 | *L. palustris*[w] | 4 | | |
| *D. versicolor* | 7 | *A. terrestris* | 4 | | |
| *L. palustris*[w] | 5 | *A. americanus* | 1 | | |
| *S. holbrookii*† | 4 | | | | |
| **J** | 0.826 | | 0.795 | | 0.794 |

Each region is sorted by the number of observations. J is the Pielou's Evenness Index. Winter calling species are indicated by '"w"'. Species unique to Florida are indicated by '†'. Species unique to South and North are indicated by '‡'. The most common species that together make up 59% or more of total observations are indicate by '*' next to their number of observations.

When regressing the number of observations against rank abundance, the exponential model over predicted expected observations of the most common species, however its performance is much better than the alternatives models (logarithmic, linear, and polynomial), which do much worse with respect to predicting the abundance of less common species.

(1) Were there significant changes in species richness over time on a given route or region?

No, with the following exceptions. <u>By Region</u>: Only two region-run combinations showed a statistically significant (0.05 alpha level) change in species richness over time, North-Winter and South-Summer (Table 9). Both were small and positive. <u>By Route</u>: The majority of route-run combinations in all regions had no significant change in species richness. Of the 16 that did, 5 routes-runs showed declines in species and 11 had increases (Table 10).

(2) Are there significant changes in species call index (as a proxy for community abundance) over time on a given route or region?

No. With the few exceptions noted in the Results section, there were no changes in ACI for the majority of region-run or route combinations examined.

(3) Do some species co-occur together?

Yes, for winter callers. No for spring and summer callers. TWINSPAN showed that winter calling species formed a group in all regions of the state. An additional species, *L. areolatus*,

**Table 6. Number of observations by region and run.**

| FLORIDA | | SOUTH | | NORTH | |
|---|---|---|---|---|---|
| RUN 1 | OBS | RUN 1 | OBS | RUN 1 | OBS |
| *P. crucifer*$^w$ | 420* | *L. sphenocephalus*$^w$ | 1189* | *P. crucifer*$^w$ | 510* |
| *P. fouquettei*$^w$ | 218* | *P. crucifer*$^w$ | 999* | *P. fouquettei*$^w$ | 383* |
| *L. sphenocephalus*$^w$ | 117 | *P. fouquettei*$^w$ | 604 | *L. sphenocephalus*$^w$ | 282 |
| *I. nebulifer* | 20 | *A. crepitans* | 60 | *A. crepitans* | 15 |
| *L. clamitans* | 5 | *L. areolatus*$^w$‡ | 10 | *L. palustris*$^w$ | 8 |
| *L. palustris*$^w$ | 4 | *L. clamitans* | 9 | *A. fowleri* | 1 |
| *D. squirellus* | 2 | *D. cinereus* | 7 | *L. clamitans* | 1 |
| *D. femoralis*† | 2 | *D. chrysoscelis* | 6 | | |
| *D. avivoca* | 1 | *L. grylio* | 4 | | |
| *L. catesbeianus* | 1 | *L. palustris*$^w$ | 3 | | |
| *L. grylio* | 1 | *L. catesbeianus* | 2 | | |
| *A. americanus* | 1 | *I. nebulifer* | 2 | | |
| | | *G. carolinensis* | 1 | | |
| | | *E. planirostris* | 1 | | |
| **J** | 0.48 | | 0.47 | | 0.60 |
| RUNS 2 AND 3 | OBS | RUNS 2 AND 3 | OBS | RUNS 2 AND 3 | OBS |
| *D. cinereus* | 222* | *A. crepitans* | 2116* | *D. chrysoscelis* | 616* |
| *A. gryllus* | 212* | *D. cinereus* | 2052* | *D. cinereus* | 546* |
| *D. chrysoscelis* | 205* | *I. nebulifer* | 1217* | *L. clamitans* | 451* |
| *L. clamitans* | 191* | *L. clamitans* | 1166* | *A. fowleri* | 259* |
| *P. crucifer* | 150* | *L. sphenocephalus*$^w$ | 1124* | *A. crepitans* | 198 |
| *I. nebulifer* | 133* | *L. catesbeianus* | 1002 | *P. fouquettei*$^w$ | 133 |
| *D. avivoca* | 110 | *D. chrysoscelis* | 827 | *P. crucifer*$^w$ | 123 |
| *L. catesbeianus* | 94 | *D. squirellus* | 770 | *L. catesbeianus* | 90 |
| *P. fouquettei*$^w$ | 90 | *A. fowleri* | 396 | *L. sphenocephalus*$^w$ | 81 |
| *L. sphenocephalus*$^w$ | 75 | *P. crucifer*$^w$ | 390 | *D. avivoca* | 76 |
| *A. crepitans* | 66 | *P. fouquettei*$^w$ | 290 | *G. carolinensis* | 52 |
| *A. americanus* | 60 | *G. carolinensis* | 276 | *D. versicolor* | 24 |
| *D. squirellus* | 53 | *L. grylio* | 247 | *D. squirellus* | 17 |
| *A. terrestris* | 41 | *D. avivoca* | 90 | *A. americanus* | 3 |
| *A. fowleri* | 38 | *D. versicolor* | 73 | *L. palustris* | 2 |
| *G. carolinensis* | 32 | *E. planirostris*‡ | 54 | *S. hurteri*‡ | 1 |
| *D. gratiosus*† | 25 | *A. gryllus* | 7 | *E. planirostris*‡ | 1 |
| *D. femoralis*† | 19 | *S. hurteri*‡ | 6 | | |
| *L. grylio* | 7 | *A. terrestris* | 4 | | |
| *A. quercicus*† | 7 | *L. areolatus*$^w$‡ | 1 | | |
| *D. versicolor* | 7 | *L. palustris*$^w$ | 1 | | |
| *S. holbrookii*† | 4 | *A. americanus* | 1 | | |
| *L. palustris*$^w$ | 1 | | | | |
| **J** | 0.87 | | 0.79 | | 0.76 |

Each Region-Run is sorted by most observed to least observed species. Observations for Runs 2 and 3 are combined. J is the Pielou's evenness index. Species unique to Florida are indicated by '†'. Species unique to South and North are by '‡'. The most common species that together make up 59% or more of total observations are indicate by '*' next to their number of observations.

**Table 7. Region-run Average Call Index (ACI) linear models.**

| Region | Run | Slope | Std. Err. | R² | p-Value |
|---|---|---|---|---|---|
| Florida | 1 | 0.01422 | 0.01485 | 0.1119 | 0.3448 |
| Florida† | 2 | -0.04938 | 0.01659 | 0.3469 | 0.0056 |
| Florida | 3 | -0.00667 | 0.01521 | 0.0885 | 0.6633 |
| North | 1 | 0.03175 | 0.01763 | 0.1710 | 0.0769 |
| North | 2 | -0.0228 | 0.01569 | 0.0034 | 0.1519 |
| North | 3 | -0.01175 | 0.01448 | 0.0002 | 0.4201 |
| South | 1 | -0.01423 | 0.00961 | 0.1599 | 0.1405 |
| South | 2 | -0.00377 | 0.00734 | 0.0550 | 0.6077 |
| South | 3 | 0.00345 | 0.00681 | 0.1643 | 0.6127 |

The model was: Region-Run ACI = Slope x (year) + intercept. †- significant ACI slope.

was present in the South region. Patterns with respect to other species were not apparent in TWINSPAN. Part of this may be due to sampling bias. More than twice as many routes in the South than for Florida and North combined were surveyed and when statewide data were pooled the groupings largely mirrored what the South TWINSPAN showed and did not provide insight for the state as a whole. Therefore, we only present the separate regional

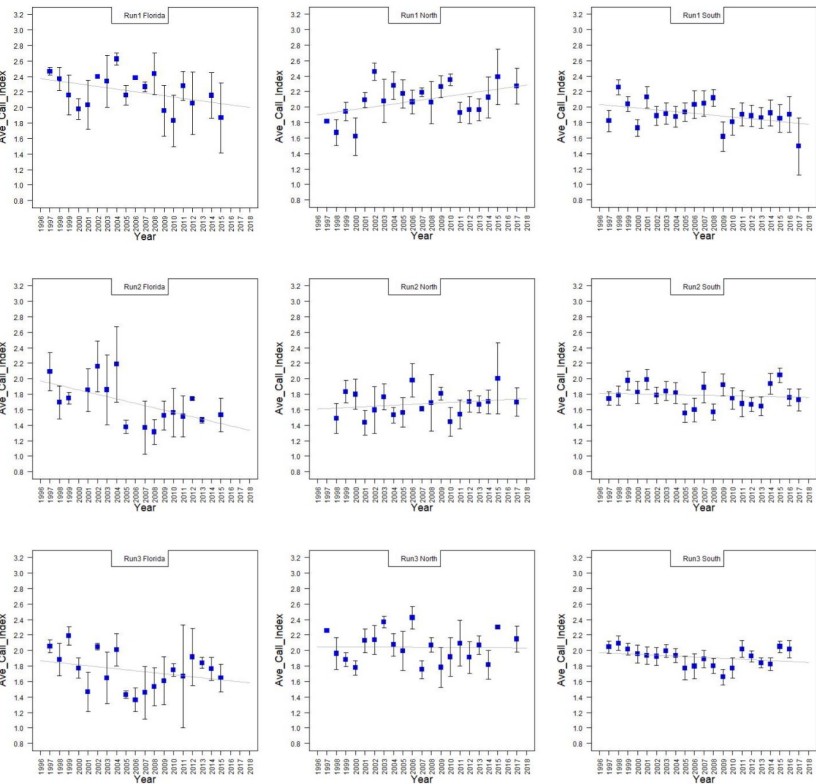

**Fig 6. Average call index by region, run and year.** The average call index (ACI) is the average of all calls documented on a given route during a given run. Top to bottom the runs are 1, 2, and 3. Right to left the columns are Florida, North, and South regions. The Y-axis is the ACI for a given year. The dots are the average of the ACI for all routes in a region in a given year with standard deviation bars. A linear regression was performed on the data. The regression models are presented in Table 7. Only Florida-Run 2 had a significant slope.

**Table 8. Route-run Average Call Index (ACI) regressions.**

| Route | Region | Run | P-value | Slope | R-square |
|---|---|---|---|---|---|
| McManus | Florida | 2 | 0.0109 | -0.0366 | 0.5764 |
| Woodland | Florida | 2 | 0.007 | -0.0516 | 0.5727 |
| Roy | North | 2 | 0.0141 | -0.0725 | 0.4071 |
| Bayou Jack | South | 1 | 0.0258 | 0.0285 | 0.3276 |
| Charenton | South | 3 | 0.0053 | -0.0267 | 0.3934 |
| Falgout Canal | South | 3 | 0.0296 | 0.0673 | 0.5734 |
| Gramercy | South | 3 | 0.0427 | -0.0442 | 0.3821 |
| Jennings | South | 2 | 0.0381 | -0.051 | 0.8078 |
| Jennings | South | 3 | 0.0179 | -0.0373 | 0.7064 |
| Montegut | South | 2 | 0.0274 | 0.45 | 0.8445 |
| Ossun | South | 3 | 0.009 | 0.0512 | 0.3384 |
| Palmetto | South | 1 | 0.0235 | 0.1196 | 0.4937 |
| Pickering | South | 3 | 0.0165 | 0.0282 | 0.3927 |
| Prairie Laurent | South | 3 | 0.03 | 0.0339 | 0.2357 |
| Price Lake | South | 3 | 0.031 | -0.0468 | 0.4731 |
| Rayne | South | 2 | 0.0047 | -0.0549 | 0.4027 |
| Rayne | South | 3 | 0.0194 | -0.0303 | 0.2968 |

Only the route-run models significant at the 0.05 alpha level and 10 or more observations are presented.

TWINSPAN results (Fig 7). However, when the TWINSPAN results for different regions were compared, no other groups besides winter callers were consistent between regions. This may be in part due to the way differences in species composition and abundances lined up between the regions (see below).

(4) Are there differences between the LAMP regions in community composition?

Yes. Run 1 had lower J indices than the scores for Runs 2 and 3 for all regions. Indeed, there was a core of 3 winter calling species (*L. sphenocephalus*, *P. fouquettei*, and *P. crucifer*) that made up more than 95% of the observations in all three regions.

Differences in the number of routes sampled and the number of runs completed make direct quantitative comparisons difficult. However, we can make a few observations. Florida had 9 species that were not found in either the South or North regions. The South and North

**Table 9. Slope of mean species richness change for region-runs from 2000 to 2016.**

| Region | Run | Number of Runs | Slope | Std. Error | t-Value | p-Value |
|---|---|---|---|---|---|---|
| Florida | 1 | 15 | 0.03152 | 0.01723 | 1.83 | 0.0758 |
| Florida | 2 | 37 | -0.03078 | 0.02055 | -1.5 | 0.1443 |
| Florida | 3 | 35 | 0.03795 | 0.03006 | 1.26 | 0.2137 |
| North† | 1 | 52 | 0.03642 | 0.01603 | 2.27 | 0.0268 |
| North | 2 | 49 | 0.02095 | 0.0196 | 1.07 | 0.29 |
| North | 3 | 50 | -0.00397 | 0.02017 | -0.2 | 0.8446 |
| South | 1 | 140 | -0.00656 | 0.00843 | -0.78 | 0.4377 |
| South | 2 | 154 | -0.00661 | 0.01296 | -0.51 | 0.6105 |
| South† | 3 | 170 | 0.04582 | 0.01569 | 2.92 | 0.0039 |

† p-Values of 0.05 or less were considered significant. All species were included and all routes-run combinations with 8 or more observations were included. Significant route-runs are indicated by '*'.

**Table 10. Mean route-run species richness linear regression.**

| Route | Region | Run | Slope | p-Value |
|---|---|---:|---:|---:|
| Loranger | Florida | 1 | 0.06306 | 0.0286 |
| Loranger | Florida | 3 | 0.20421 | 0.0072 |
| Woodland | Florida | 2 | -0.06768 | 0.032 |
| Cotton Valley | North | 1 | 0.07749 | 0.0082 |
| Rocky Branch | North | 1 | -0.0823 | 0.0099 |
| McManus | North | 2 | -0.04047 | 0.0098 |
| Roy | North | 2 | -0.07179 | 0.0484 |
| Bayou Jack | South | 3 | 0.1054 | 0.0458 |
| Bayou Sorrel | South | 3 | 0.14544 | 0.044 |
| Gramercy | South | 3 | 0.13369 | 0.044 |
| Ossun | South | 1 | 0.01539 | 0.0437 |
| Ossun | South | 2 | 0.03273 | 0.0044 |
| Ossun | South | 3 | 0.08649 | 0.0464 |
| Pickering | South | 3 | 0.06653 | 0.0198 |
| Rayne | South | 2 | -0.04549 | 0.0298 |
| Rayne | South | 3 | 0.06471 | 0.0454 |

Over 200 combinations of route-run were modeled. Only models with p-Values $\leq 0.05$ and 8 or more observations are presented. All species were included in the species richness calculations.

regions together had 3 species not found in Florida. Therefore, based on comparison of the species list, we'd group South and North together, with Florida in a different group.

It should be noted that the species that did set the regions apart were relatively rare (Table 6). The uniquely Florida species only accounted for 2.16% of the relative abundances for that region, and the species that were not found in Florida only accounted for 0.48% and 0.05% of the relative abundances in South and North respectively.

When we compare the most abundant species however, a different picture emerges (Table 6). Of the 6 most common species in each region, North shares 5 of its 6 with Florida, while South shares only 3 each with Florida and North. When we look at the highest ranked species (those that make up 59% or more of the observations), all 4 of North's species are shared with Florida, but only 2 of its 4 species are shared with South. So, while comparisons of the species list would group South and North, comparisons of actual observations would group North and Florida. A comparison of the region-runs finds the same pattern, whereby the lists of the most observed species in North and Florida, by rank or relative abundance, are more similar to each other than either is to South. This is true for both Run 1 and Runs 2 and 3 combined.

In summary, the regions and runs do differ. Winter calling species held up as a community in and of themselves both in composition and relative abundances. If we were simply comparing species lists, the North would be seen as a subset of the South. When comparing abundances however, North and Florida are more similar to each other than either is to South both regionally and seasonally.

## Species level questions

1. Which species were detected during the surveys, when and where?
   Twenty-six species were detected (Table 4). Five were classified as winter callers, 2 spring callers, 2 were equally divided between spring and summer, and the rest (17) were summer

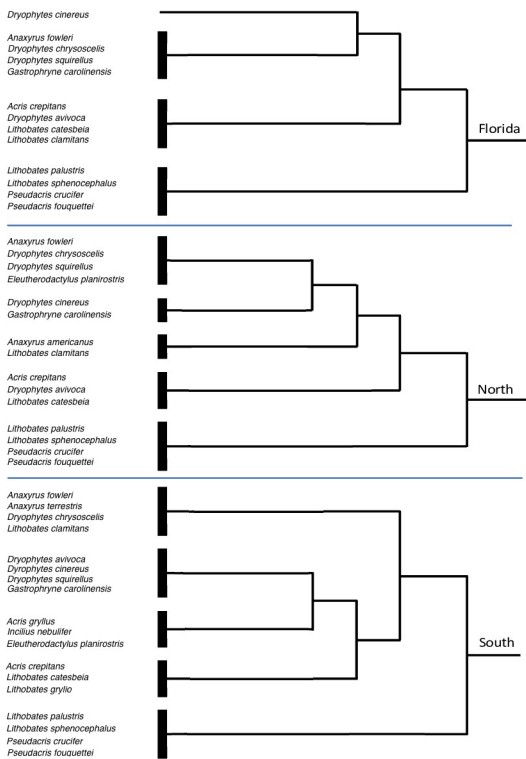

**Fig 7. Two Way Indicator Species Analysis of the 15 most common species for the LAMP regions.** Winter calling species (*L. palustris*, *P. crucifer*, *P. fouquettei*, and *L. sphenocephalus*) are the first species to cluster together. No other groupings were consistent across the regions.

callers. No species were found outside of the regions for which they were already known. Three native species were not observed, but the survey routes did not overlap with their known distributions and two are considered extirpated since the 1960's. Two recently introduced species, *O. septentrionalis* and *E. cystignathoides*, were not observed along any survey route. This may be due to fact that their distributions may be in areas without survey routes.

2. Did any species show a change in the frequency of observations or abundances for a given region, route or run?
Yes they did, but no consistent pattern emerged for any given species across all routes in a given region or statewide. The sACI for a species along a given route-run was not related to how often a species was observed along the same route (Fig 4). The rarest species in the LAMP data, *S. holbrookii*, had the highest sACI. Eleven (11) of the 15 most common species had at least one GAM with significant changes in how often they were observed along a route over time (Table 11). All but one model had at least one change in slope sign from positive to negative or negative to positive. Seven (7) models showed an overall increase and 1 a decrease in percent observation over time. The majority of GAM models of percentage of stops where a species was observed on a route-run were not significant, and those that were in general had multimodal patterns.

3. Are there differences in LAMP region community composition?
Yes there were. The regions had different species richness values: Florida-23; South- 22; North- 17. Frogs that called during the winter also called during the summer, but 2/3 of the

**Table 11. Route-run and region-run species percent observation GAM results.**

| Species | Region | Route | Run | Slope Sign Changes | Pattern |
|---|---|---|---|---|---|
| *L. catesbeianus* | South | Bayou Jack | 3 | 3 | Multimodal |
| *L. clamitans* | Florida | Loranger | 3 | 1 | Convex-Increasing |
| *L. clamitans* | Florida | REGION | 3 | 3 | Multimodal |
| *L. clamitans* | Florida | Woodland | 3 | 3 | Multimodal |
| *L. clamitans* | North | REGION | 2 | 2 | Multimodal |
| *L. clamitans* | North | Cotton Valley | 2 | 2 | Multimodal |
| *L. clamitans* | North | Koran | 2 | 3 | Multimodal |
| *L. clamitans* | North | REGION | 3 | 1 | Convex |
| *L. clamitans* | North | Cotton Valley | 3 | 2 | Multimodal |
| *L. clamitans* | South | Gramercy | 3 | 1 | Convex-Increasing |
| *L. clamitans* | South | Pickering | 3 | 2 | Multimodal |
| *P. fouquettei* | North | Roy | 1 | 1 | Concave |
| *P. fouquettei* | South | REGION | 1 | 1 | Concave |
| *P. fouquettei* | South | Bayou Jack | 1 | 1 | Concave |
| *D. chrysoscelis* | Florida | McManus | 2 | 0 | Increasing |
| *D. chrysoscelis* | Florida | REGION | 3 | 1 | Convex |
| *D. chrysoscelis* | Florida | Woodland | 3 | 1 | Convex |
| *G. carolinensis* | South | REGION | 3 | 2 | Multimodal |
| *D. cinereus* | South | Bayou Jack | 3 | 2 | Multimodal-Increasing |
| *I. nebulifer* | South | Prairie Laurent | 3 | 2 | Concave |
| *I. nebulifer* | South | Bayou Sorrel | 3 | 2 | Multimodal/-Increasing |
| *A. crepitans* | North | Cotton Valley | 2 | 1 | Concave |
| *A. crepitans* | South | Egan | 2 | 2 | Multimodal |
| *A. gryllus* | Florida | REGION | 2 | 3 | Multimodal |
| *A. gryllus* | Florida | Loranger | 2 | 2 | Multimodal |
| *A. gryllus* | Florida | McManus | 2 | 3 | Multimodal |
| *A. gryllus* | Florida | Loranger | 3 | 3 | Multimodal |
| *A. gryllus* | Florida | McManus | 3 | 3 | Multimodal |
| *L. sphenocephalus* | Florida | Woodland | 1 | 2 | Multimodal |
| *L. sphenocephalus* | North | Roy | 1 | 1 | Convex |
| *L. sphenocephalus* | South | REGION | 1 | 1 | Concave |
| *L. sphenocephalus* | South | Pickering | 1 | 3 | Multimodal |
| *L. sphenocephalus* | South | Prairie Laurent | 1 | 2 | Multimodal-Decreasing |
| *P. crucifer* | North | REGION | 1 | 1 | Concave |
| *P. crucifer* | North | Koran | 1 | 1 | Concave |
| *P. crucifer* | North | Roy | 1 | 1 | Concave-Increasing |
| *P. crucifer* | North | Cotton Valley | 1 | 1 | Increasing |
| *D. squirellus* | South | Brannon | 3 | 2 | Multimodal |
| *D. squirellus* | South | Rayne | 3 | 1 | Multimodal |

Only models with significant p-values ($\leq 0.05$) and 10 or more observations are presented. Within a species they are grouped by Region, Route and Run, and the number of changes in the sign (positive or negative) of the slope. GAM model run plots are in S1 File. See Fig 1 in S1 File for examples of the shapes. Multimodal-Increasing means that the pattern was multimodal but with a significant increase overall.

summer or spring callers did not call during the winter. Consequently, the species richness for Run 1 was lower: South-14, Florida-12, and North -7. The Florida region had species that were restricted to east of the Mississippi River. Based on species lists, North and South

grouped together. Based on the mix of species observations Florida and North grouped together. Only winter callers held together as a community with TWINSPAN.

4. What are the most common species?

*Lithobates sphenocephalus* and *P. crucifer* were the most commonly observed species. The most observed species statewide was *L. sphenocephalus*. When examined by region the most common species in the South was *L. sphenocephalus* and the most common species in Florida and North were *P. crucifer*. Both are winter callers. The species with the highest sACI, *S. holbrookii*, was also the species with the fewest observations suggesting that for some species, the call index may not be a good measure of abundance.

## Previous studies

Villena et al. [19] used a portion of the LAMP data as part of a larger study of frog population trends in the Southeast. Their study used fewer species and covered fewer years than the data used for this study. They used occupancy modeling to document changes in their occupancy. In Louisiana only two toad species, Fowler's Toad (*Anaxyrus fowleri*) and Southern Toad (*Anaxyrus terrestris*) were found to have significant changes in their occupancy slopes (positive for Fowler's, negative for Southern). The other species were found to have slopes with confidence intervals that overlapped with 0 (no change). In our analyses, we found *A. fowleri* to have a multimodal abundance pattern. We did not conduct an analysis of the *A. terrestris* data because there were not enough observations (N = 45) to meet our inclusion criteria (>200 observations). In this study species patterns varied between regions. *A. fowleri* made up less than 4% of the observations for Florida and South but 11.86% of the observations for North. The only significant GAM for *A. fowleri* was for Rocky Branch (North). It included only 6 years and was multimodal.

In this study, 14 species were found to have significant GAM models with respect to changes in observations for 71 different route- run combinations (Table 8). The relevance of this is the majority of species had multiple peaks or dips in their populations over the study period and that short observation windows of less than 10 years are probably inadequate to document long-term trends.

A study in the Atchafalaya Basin of Louisiana [20] found that all 12 species detected in the study experienced declines in occupancy during their 5-year observation period (2002–2006). Our GAM analysis of percent observations along a route (Table 11) suggests that their observations may not be generalizable for the rest of Louisiana, and that their observation window may have been too short to assess long-term species status.

## Conclusions

Taken together, our results lead to three broad conclusions. (1) Can call monitoring detect trends in frog populations? Yes. We were able to follow trends in abundance and distribution for some routes for 20 years. During this time populations often went through one or more cycles of changes in abundance. The length of these cycles demonstrated the importance of long-term (10+ years) monitoring. Observation periods of less than 10 years, or with long multiyear gaps, are likely to be too short and/or incomplete to determine if a species is experiencing long-term declines or increases in abundance or occupancy. (2) Are frog abundances in Louisiana declining? Perhaps in some locations, but overall no. In some cases abundances increased. Our analyses detected no long-term trends in either the percent observation along a given route or region, or ACI along a given route or region. We did not detect long-term species richness trends in any of the three LAMP survey regions. (3) Are their regional trends in

frog abundance, and are frog populations in a given region synchronous? With a couple exceptions, no to both questions. The routes were, in general, not synchronized with respect to their patterns of species richness, abundance (call index), and species occupancy (percent presence) and different regions had different dominant species.

New threats to frog species have emerged over the last 10 years. *O. septentrionalis*, has recently invaded the cities of New Orleans and Lafayette, and have been expanding their range from urban areas [21]. They are predators on other species of frogs [21]. A second potential emerging threat are Giant Apple Snails (*Pomacea maculata*), which are potential frog egg predators [22]. *Pomacea maculata* have been expanding their range throughout the southern parishes of Louisiana [23] and are often found in many of the same habitats that frogs occupy. Continued monitoring my help inform whether these or other unknown threats are impacting frog populations. Monitoring provides an important baseline for future assessments. The LAMP data provides a baseline that can be used to determine if there are significant changes in frog populations or distributions caused by these or other threats.

## Supporting information

**S1 Table. LAMP routes.** Route name, NAAMP number, latitude and longitude of first stop. (DOCX)

**S1 File. S1-S21 Figs.** GAM plots of the percentage of stops a species was observed calling along a given route-run versus year.
(ZIP)

## Acknowledgments

We would like to thank the many LAMP volunteers who put in many years and drove 1,000's of kilometers to collect the data used here. We would like to thank the Louisiana Department of Wildlife and Fisheries Natural Heritage Program for providing the survey data. We would like to thank Stephen A. Hartley for providing us with a revised LAMP route map. We would like to thank the editor (J. Bossart) and the reviewers (C.K. Beachy, K.R. Messenger, H. Waddle, and anonymous) for their comments and suggestions.

The U.S. Geological Survey's Ecosystems Mission Area provided support for this analysis. Any use of trade, firm, or product names is for descriptive purposes only and does not imply endorsement by the United States Government.

## Author Contributions

**Conceptualization:** Jacoby Carter.

**Data curation:** Jacoby Carter, Jeff Boundy.

**Formal analysis:** Jacoby Carter, Darren Johnson.

**Methodology:** Jacoby Carter, Darren Johnson.

**Project administration:** Jacoby Carter, Jeff Boundy.

**Validation:** Jeff Boundy.

**Writing – original draft:** Jacoby Carter, Darren Johnson, William Vermillion.

**Writing – review & editing:** Jacoby Carter, Darren Johnson, Jeff Boundy, William Vermillion.

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
