## [Decision Letter · Decision Letter 0]

16 Sep 2020

PONE-D-20-21124

The Louisiana Amphibian Monitoring Program from 1997 to 2017: Results, Analyses, and Lessons Learned.

PLOS ONE

Dear Dr. Carter,

Thank you for submitting your manuscript to PLOS ONE. After careful consideration, we feel that it has merit but does not fully meet PLOS ONE’s publication criteria as it currently stands. Therefore, we invite you to submit a revised version of the manuscript that addresses the points raised during the review process.

Please submit your revised manuscript within 30 days. If you will need more time than this to complete your revisions, please reply to this message or contact the journal office at plosone@plos.org. Please include the following items when submitting your revised manuscript:

We look forward to receiving your revised manuscript.

Kind regards,

Yong Zhang

Academic Editor

PLOS ONE

Additional Editor Comments:

I have received two reviewers comments. Both of them think it is a nice manuscript with only a few comments. I found those comments are not difficult to be addressed and hence suggest a Minor Revision.

Journal Requirements:

2. In your Methods section, please provide additional location information of the study sites, including geographic coordinates for the data set if available.

3. In your Methods section, please provide additional information regarding the permits you obtained for the work. Please ensure you have included the full name of the authority that approved the study sites access and, if no permits were required, a brief statement explaining why.

5. We note that Figure 1 in your submission contain map images which may be copyrighted. All PLOS content is published under the Creative Commons Attribution License (CC BY 4.0), which means that the manuscript, images, and Supporting Information files will be freely available online, and any third party is permitted to access, download, copy, distribute, and use these materials in any way, even commercially, with proper attribution. For these reasons, we cannot publish previously copyrighted maps or satellite images created using proprietary data, such as Google software (Google Maps, Street View, and Earth). For more information, see our copyright guidelines: http://journals.plos.org/plosone/s/licenses-and-copyright.

5.1.    You may seek permission from the original copyright holder of Figure 1 to publish the content specifically under the CC BY 4.0 license. 

5.2.    If you are unable to obtain permission from the original copyright holder to publish these figures under the CC BY 4.0 license or if the copyright holder’s requirements are incompatible with the CC BY 4.0 license, please either i) remove the figure or ii) supply a replacement figure that complies with the CC BY 4.0 license. Please check copyright information on all replacement figures and update the figure caption with source information. If applicable, please specify in the figure caption text when a figure is similar but not identical to the original image and is therefore for illustrative purposes only.

Reviewers' comments:

Reviewer's Responses to Questions

**Comments to the Author**

1. Is the manuscript technically sound, and do the data support the conclusions?

Reviewer #1: Partly

Reviewer #2: Yes

2. Has the statistical analysis been performed appropriately and rigorously? 

Reviewer #1: Yes

Reviewer #2: Yes

3. Have the authors made all data underlying the findings in their manuscript fully available?

Reviewer #1: Yes

Reviewer #2: Yes

4. Is the manuscript presented in an intelligible fashion and written in standard English?

Reviewer #1: Yes

Reviewer #2: Yes

5. Review Comments to the Author

Reviewer #1: This study reviewed and analyzed frog call data from LAMP and evaluated the species richness, diversity and distributions, population dynamics of the three survey regions in Louisiana. The authors found the frog community of North region is in the middle between the South and Florida parishes. The authors also concluded monitoring for more than 10 years is vital to assess population and occupancy trends.

For amphibians, 10 years are usually enough to evaluate the population dynamics. Therefore, my major concern is the survey method to count the abundance in this study. Has been it test the correlation between call indexes and species abundance? The frog call activities can largely fluctuate when ambient temperature or breeding stages (especially important for explosive breeding species) vary. Were these variables added to the statistic models? In addition, did these surveys complete at daytime or at night or both? I strongly suggest the authors discuss the limitation of the methods when draw a conclusion about the time scale for amphibian population monitoring.

Reviewer #2: Overall, most everything was fine with few errors that I caught:

line 130: "of" prior to "west" can be deleted.

line 139: There's a strange placement of "nor" at the end of a sentence. "...did not need permits nor." Reword.

line 144: Just remove the miles and stick with km. No scientific article should be using miles.

line 157: I think your "(1)" is supposed to be an "(A)"

line 263: "Were" not "Where"

line 449: Remove "it" in "...of runs completed make it direct..."

line 494: "where" not "were"

line 519: sci name not in italics

line 520: sci name not in italics

Perhaps it is just me, but I had to read lines 155-156 a few times. Initially, I thought if condition B is satisfied, then A is automatically satisfied, in which case listing A is null. Eventually I got it. If other reviewers perhaps had a similar issue, needing to read this sentence a few times, then I'd suggest maybe providing an example. Or perhaps suggesting B first, and then A. "...routes had to have: (A) Surveys need to span 20 or more years, and (B) within that time span, there need to be data sets from 8 or more different years." I think it was the small (8) to large (20) that kind of threw me. So, I would suggest rewording to make the sentence/ conditions a little more clear, if other reviewers noted this area as being a bit awkward. If I'm the only one, then disregard.

6. PLOS authors have the option to publish the peer review history of their article (what does this mean?). If published, this will include your full peer review and any attached files.

Reviewer #1: No

Reviewer #2: **Yes: **Kevin R Messenger

---

## [Author Response · Author response to Decision Letter 0]

3 Mar 2021

We agreed with most of the changes the reviewers suggested. Our letter of reconciliation and response outlines the changes we made.

---

## [Decision Letter · Decision Letter 1]

5 May 2021

PONE-D-20-21124R1

The Louisiana Amphibian Monitoring Program from 1997 to 2017: Results, Analyses, and Lessons Learned.

PLOS ONE

Dear Dr. Carter,

Thank you for submitting your manuscript to PLOS ONE. After careful consideration, we feel that it has merit but does not fully meet PLOS ONE’s publication criteria as it currently stands. Therefore, we invite you to submit a revised version of the manuscript that addresses the points raised during the review process.

Although I'm keeping the editorial decision as Minor Revision (the data and analyses are largely solid), the Introduction and Discussion, nonetheless, need some solid reworking.  As reviewer 3 noted, neither of these sections are well-developed.  I fully agree.  PLoS ONE is an international journal and the Introduction needs to place the study into some broader context rather than simply describing a program specific to a single state in the United States.  For example, what was the motivation for the study?  In what way will the results be relevant?  Why would someone care to read your publication?  Also, although publications commonly include goals/objectives/predictions towards the end of the Introduction, these are woven within the final 1 or 2 paragraphs of that section.  I don't know that I have ever seen a publication where these have been simply numerically listed.  Ditto the Discussion.  As it currently stands, most of this section is just restating results. The Discussion section needs to be a developed discussion of those results.  The citations below are recent PLoS One publications in the same general topic area and are provided as representative examples of what a typical publication might look like.  

Hutto Jr, D., & Barrett, K. (2021). Do urban open spaces provide refugia for frogs in urban environments?. *PloS one*, *16*(1), e0244932. 

Rivas, G. A., et al. (2021). Biogeographical patterns of amphibians and reptiles in the northernmost coastal montane complex of South America. *PloS one*, *16*(3), e0246829.

Ceron, K., Santana, D. J., & Valente-Neto, F. (2020). Seasonal patterns of ecological uniqueness of anuran metacommunities along different ecoregions in Western Brazil. *PloS one*, *15*(9), e0239874.

All reviewers have had favorable comments about the overall quality of your data and statistical approaches used.  I look forward to receiving a revised version, where the Introduction and Discussion and have been better fleshed out.

We look forward to receiving your revised manuscript.

Kind regards,

Janice L. Bossart

Academic Editor

PLOS ONE

Journal Requirements:

Additional Editor Comments (if provided):

Reviewers' comments:

Reviewer's Responses to Questions

**Comments to the Author**

1. If the authors have adequately addressed your comments raised in a previous round of review and you feel that this manuscript is now acceptable for publication, you may indicate that here to bypass the “Comments to the Author” section, enter your conflict of interest statement in the “Confidential to Editor” section, and submit your "Accept" recommendation.

Reviewer #2: All comments have been addressed

Reviewer #3: (No Response)

2. Is the manuscript technically sound, and do the data support the conclusions?

Reviewer #2: Yes

Reviewer #3: Yes

3. Has the statistical analysis been performed appropriately and rigorously? 

Reviewer #2: Yes

Reviewer #3: Yes

4. Have the authors made all data underlying the findings in their manuscript fully available?

Reviewer #2: Yes

Reviewer #3: Yes

5. Is the manuscript presented in an intelligible fashion and written in standard English?

Reviewer #2: Yes

Reviewer #3: Yes

6. Review Comments to the Author

Reviewer #2: As I did not have many issues with the first draft, and other reviewers had more - this version is more than acceptable to me.

Reviewer #3: My overall impression is that I am impressed with this massive project and the data. Very important that they are presented. The analysis are appropriate and are communicated clearly.

I would wish for more of a proper discussion section. Except for ll. 542-550, the current version just reads as a rehash of the Results. Even though I personally don’t find call surveys my cup of tea, this is such a large effort and such a large data set, that any mechanical problems that would usually be inherent in a report like this one are simply overrun by the enormity of the data. So, I’m a hater, but I love the data and effort reported in this manuscript and hope to read it in print soon.

(This is my first view of this manuscript; many of the parts read like they are responses to previous reviewers and so feel a little clunky. I've mentioned these points below).

Other comments intended to help improve the writing:

Can the volunteers willing to be named be included? This huge project is a great opportunity to recognize them.

31 species? There are 32 reported species in Louisiana.

l. 55 - ‘synchronicity’ is unclear

Make the abstract a single paragraph.

l. 64 - remove ‘perceived’ and ‘North American’

Is this really an Introduction? It reads as a description of the project. It is not placed in the context of the global question of amphibian decline and the monitoring programs put in place to address this problem.

ll. 69-72 and 77-79 are better placed in Methods

l. 80. move [1] before period.

ll. 92-93 - this is not a novel question; species co-occurences and associated phenology and habitat are already known and described in the literature. And this study is not the best way to address this.

ll. 98-99 - see previous.

ll. 113-116 and l. 119 - The Florida Parishes make sense because it’s a biogeographic thing. But do better at explaining the choice of 31 degrees. I think it’s a simply reason, but write it. At this point, it reads as something arbitrary but I doubt that it was chosen arbitrarily.

TABLE 1 - I understand why there might be a wide range of variation in number of stops. But try to describe why.

l. 161 (and elsewhere) - use your term ‘observers’ not ‘surveyors’

l.177 - data ‘are’; replace ‘is’

l. 181 - see previous

l.209-221 - remove the descriptions of species indices calculations; these are common knowledge

The GAM is description is critical to this manuscript and it’s clear. Nice job!

ll. 244-248 and Fig. 2. - this section feels like something that may have been requested by an earlier reviewer. I find it unnecessary; it’s intuitive and use of the GAM analyses are fairly common. A bit like explaining how a regression works.

l. 261-262 - I’m interested! Why were there fewer routes?

l. 268 - I LOVE the numbers 54 and (especially) 12,792! This line represents how much work has been put into this project and why I think some place for the volunteers names would improve this manuscript.

l.272 - 11 species calling at a single stop! That is neat and impressive and shows how good the observers were.

l. 278-284 - I don’t understand the reason to include this.

Fig. 5 - This could be deleted and the regression statistics simply incorporated into the text.

ll. 306-314 and elsewhere - Taxonomy matters in some journals and I’ll assume that it does in PLOS ONE. It is not “spring peepers”; it is Spring Peepers; it is not “Southern (Acris gryllus) and northern cricket frogs (Acris crepitans)”; it is Southern Cricket Frogs (Acris gryllus) and Northern Cricket Frogs (Acris crepitans).

In addition, at present, the preferred use is Hyla not Dryophytes (along with the associated gender changes in species epithets. See Crother et al for current taxonomy

ll. 319-322 - I think this is an excellent place for proper digging. Why not compare to older patterns? For example, compare with Fig. 6 in Dundee and Rossman. This would allow one to address the question of changing call phenology associated with a 35 year change in climate.

7. PLOS authors have the option to publish the peer review history of their article (what does this mean?). If published, this will include your full peer review and any attached files.

Reviewer #2: No

Reviewer #3: **Yes: **Christopher K. Beachy

---

## [Author Response · Author response to Decision Letter 1]

28 Jun 2021

To: Janice L. Bossart, Academic Editor

Date: Monday, June 28, 2021

Re: Reconciliation Letter for PONE-D-20-21124R1. 

The Louisiana Amphibian Monitoring Program from 1997 to 2017: Results, Analyses, and Lessons Learned.

Dear Ms. Bossart,

I have revised the above manuscript as per your, and the reviewer’s comments. I appreciate the extra time you gave me (July 2) to complete my revisions and response. Included in the resubmitted package are a copy of the manuscript with the track changes from the last submission, a 'clean' copy of the manuscript, and a letter of reconciliation.

In order to make certain I didn’t miss any comments/suggestions/changes asked, I used the May 5 letter as the base document in the letter of reconciliation. Each issue in your May 5 letter is followed by an outline of the I made, rebuttal or other response. To help separate my responses from those of yourself or the reviewers, I used a different font, Times New Roman for my responses in the letter of reconciliation. 

Some of the changes I made were as follows:

Introduction. I rearranged the sentences in the introduction to address this problem as to ‘Why?’ of the study. While it was there in the first sentence, (perceived amphibian declines), it got lost in the details that followed. I also rewrote a few sentences to clarify which questions we were answering with LAMP data. Finally, I removed the reference to a previous study in the introduction. The reference to the previous study broke the flow of the introduction and really didn’t add to the set up for the Introduction. I removed sentences that were better left to the Methods section.

Discussion Structure. Here we focused on the community level and species level questions. There were many different analytical approaches we used with the LAMP data to answer the multiple questions. In some cases more that one approach was used to answer a question. 

We used the discussion to tie what was presented in the Results section directly back to the questions we asked at the end of the Introduction. To make this section cleaner, after restating the question, I provide a direct, simple answer. I then follow up with a paragraph that abstracts from the results. I removed as detailed recapitulation from the results that I thought I could. 

Conclusion. The conclusion section now focuses on the three broad questions. “Can this method be used to monitor for changes in frog abundance or distribution?” “Did we see changes in abundance or frog distributions?” “Are their regional differences in frog distribution and abundance?” Followed by a section on why continued monitoring is important.

Additional Changes. 

• In order to distinguish between average call index (ACI) for all species in a given region/route/run combination and those for just one species in a given region/route/run combination in the body of the manuscript I now use ‘ACI’ when referring to all species on a given region/route/run and ‘sACI’ when referring to just one species on a given region/route/run. This should remove any ambiguity. 

I have no changes in my financial statements, data repository information, or standard operating procedures since our last submission.

I have reviewed our reference list, it is correct and only references needed are included.

Reviewer Comments.

Reviewer 2 had no changes or comments. S/He stated that they were satisfied with the revised manuscript as is.

Reviewer 3 was a new reviewer. He had some specific questions and suggestions. They are covered in detail in the attached letter of reconciliation date June 28. In general I accepted his style suggestions and revised some things to make them clearer. Some suggestions I did not use for reasons provided in the letter of reconciliation. 

Editor comments. I reorganized the introduction, discussion, and conclusions to address the issues raised. Please see the attached letter of reconciliation for details.

---

## [Decision Letter · Decision Letter 2]

16 Jul 2021

PONE-D-20-21124R2

The Louisiana Amphibian Monitoring Program from 1997 to 2017: Results, Analyses, and Lessons Learned.

PLOS ONE

Dear Dr. Carter,

Thank you for submitting your manuscript to PLOS ONE. After careful consideration, we feel that it has merit but does not fully meet PLOS ONE’s publication criteria as it currently stands. Therefore, we invite you to submit a revised version of the manuscript that addresses the points raised during the review process.

Before your manuscript can be accepted, a few small changes are needed and numerous errors need to be corrected.  These are listed below.  Line numbers refer to the clean copy.

Move Fig. 2 to Supplemental Material.  Re-number all Figs to reflect this change and modify text in all places where former Fig. 2 is referenced.Table 1 & 9 have small alignment issues.  Table 1: Spacing of the title 'Route Name' on the right side needs moved left to correspond with the placement of that same title on the left side.  Table 9: footnote... align on left with no indent.  Check that all information in all Tables is formatted consistently.Formatting of references is inconsistent, e.g. authors followed by periods or commas, initials set off by commas or not. Carefully check that all are in agreement and in the format used by PLoS One.Lines 81-83. #3 Wording is awkward. Reword. I suggest: If there are trends, are these associated  with frog communities as a whole or with individual species or with both.Line 89. Missing 'they'Line 111.  'The state was...' Line 117. 'where' not 'were'Line 122.  'were' not 'was'...data wereLine 134. 'between at' makes no sense. Delete 'between'Lines 139-140. Multiple grammar issues. Replace with: ArcInfo was used to superimpose on the public domain base map the locations of NAAMP route survey routes and then to generate a TIFF file of the new map. Line 146. Delete 'stops'Line 152. 'require' not 'required'Lines 178 & 180. Replace 'it is' with 'data are' Line 336. Comma after 'North'Line 340. Add 'indicated... are indicated byLine 349. Delete 'a'Line 403. Replace 'between the' with 'across'Line 413. 'more' not 'move'Line 415. Delete 'the'.. Change Fig 2 reference to reflect move to supplemental Line 430. 'which' should be preceded by a commaLine 434. 'combinations' not 'combination'Line 437. Replace 'had' with 'showed'Lines 479-480. The correct phrase is 'in and of themselves' Line 564. 'which' should be preceded by a commaLine 584. Delete extra space before SauerLines 329, 331, 516, 532, 561 & 564. Spell out genus at start of a sentence.

We look forward to receiving your revised manuscript.

Kind regards,

Dr. Janice L. Bossart

Academic Editor

PLOS ONE

Journal Requirements:

Reviewers' comments:

Reviewer's Responses to Questions

**Comments to the Author**

1. If the authors have adequately addressed your comments raised in a previous round of review and you feel that this manuscript is now acceptable for publication, you may indicate that here to bypass the “Comments to the Author” section, enter your conflict of interest statement in the “Confidential to Editor” section, and submit your "Accept" recommendation.

Reviewer #3: All comments have been addressed

2. Is the manuscript technically sound, and do the data support the conclusions?

Reviewer #3: Yes

3. Has the statistical analysis been performed appropriately and rigorously? 

Reviewer #3: Yes

4. Have the authors made all data underlying the findings in their manuscript fully available?

Reviewer #3: Yes

5. Is the manuscript presented in an intelligible fashion and written in standard English?

Reviewer #3: Yes

6. Review Comments to the Author

Reviewer #3: The authors have responded to all the suggestions that I provided in my first review. Where they have not adopted my suggestions, they have adequately described why. I'm good with what they have adopted and what they have not.

7. PLOS authors have the option to publish the peer review history of their article (what does this mean?). If published, this will include your full peer review and any attached files.

Reviewer #3: **Yes: **Christopher K. Beachy

---

## [Author Response · Author response to Decision Letter 2]

21 Jul 2021

Dear Dr. Bossart, 

Thank you. Your suggestions and directions were clear.

You wrote:

“Before your manuscript can be accepted, a few small changes are needed and numerous errors need to be corrected. These are listed below. Line numbers refer to the clean copy.” Followed by a list of required changes. I accepted all of your suggestions. Please find below point by point how they were addressed.

• Move Fig. 2 to Supplemental Material. 

 RESPONSE: Made the suggested change. I re-number all figures to reflect this change and modify text in all places where former Fig. 2 is referenced.

• Table 1 & 9 have small alignment issues. Table 1: Spacing of the title 'Route Name' on the right side needs moved left to correspond with the placement of that same title on the left side. Table 9: footnote... align on left with no indent. Check that all information in all Tables is formatted consistently.

 RESPONSE: All tables are now in 12-point font, all columns with text are left justified, all columns with numbers are center justified. I fixed the alignment the Table 9 footnote as directed.

• Formatting of references is inconsistent, e.g. authors followed by periods or commas, initials set off by commas or not. Carefully check that all are in agreement and in the format used by PLoS One.

 RESPONSE: After checking your submission guidelines (https://journals.plos.org/plosone/s/submission-guidelines#loc-references), I went through the reference section and made corrections. Author initials, are set off by commas unless they are the last author listed. The last author is separated from the journal title with a period. Additional changes made were: (1) urls with periods at the end that were not part of the referenced url were removed as indicated in the examples; (2) the year of publication for one software package was moved to the end of the reference so it was consistent with the other references.

• Lines 81-83. #3 Wording is awkward. Reword. I suggest: If there are trends, are these associated with frog communities as a whole or with individual species or with both. 

 RESPONSE: Used suggested wording

• Line 89. Missing 'they' Done

• Line 111. 'The state was...' Done

• Line 117. 'where' not 'were' Done

• Line 122. 'were' not 'was'...data were Done

• Line 134. 'between at' makes no sense. Delete 'between' Done

• Lines 139-140. Multiple grammar issues. Replace with: “ArcInfo was used to superimpose on the public domain base map the locations of NAAMP route survey routes and then to generate a TIFF file of the new map. “

 RESPONSE: Used suggested wording

• Line 146. Delete 'stops' Done

• Line 152. 'require' not 'required' Done

• Lines 178 & 180. Replace 'it is' with 'data are' Done

• Line 336. Comma after 'North' Done

• Line 340. Add 'indicated... are indicated by Done

• Line 349. Delete 'a' Done

• Line 403. Replace 'between the' with 'across' Done

• Line 413. 'more' not 'move' Done

• Line 415. Delete 'the'.. Change Fig 2 reference to reflect move to supplemental Done

• Line 430. 'which' should be preceded by a comma Done

• Line 434. 'combinations' not 'combination' Done

• Line 437. Replace 'had' with 'showed' Done

• Lines 479-480. The correct phrase is 'in and of themselves' Done

• Line 564. 'which' should be preceded by a comma Done

• Line 584. Delete extra space before Sauer Done

• Lines 329, 331, 516, 532, 561 & 564. Spell out genus at start of a sentence.“

 RESPONSE: Correction made. In addition to the lines given, I went through the manuscript and found additional places where that change needed to be made and made them.

OTHER CHANGES

(1) I ran the manuscript through the spelling/grammar checker to catch any new errors that might have crept in during this most recent editing process. There were a few suggested changes that I accepted.

(2) I removed the USGS required statement that the manuscript is ‘pre-decisional’. It has been approved for publication.

---

## [Editor Report · Decision Letter 3]

29 Jul 2021

PONE-D-20-21124R3

The Louisiana Amphibian Monitoring Program from 1997 to 2017: Results, Analyses, and Lessons Learned.

PLOS ONE

Dear Dr. Carter,

Thank you for submitting your manuscript to PLOS ONE. After careful consideration, we feel that it has merit but does not fully meet PLOS ONE’s publication criteria as it currently stands. Therefore, we invite you to submit a revised version of the manuscript that addresses the points raised during the review process.

I'm sorry I need to send the manuscript back to you again, but PLoS One manuscripts are not copy edited.  Once I accept it will go to the production staff.  Just a few things:

Line 118 (Clean copy), i.e. Table 1 footnote.  Although you indicated 'was' had been changed to 'were', it hasn't.  Please change, '...data from that route were...'Table 1.  Please ensure the entire Table can be see in Draft view in Word since it extends past the right margin.  I can't doublecheck as I only receive a pdf.Scientific nomenclature.  I appreciate your effort, but genus is generally only spelled out on first use in the main text and at the start of a sentence and abbreviated otherwise.  Sorry, I should have caught this issue earlier, but it only hit me when even more were spelled out in this version.  In the main text, spell out genus once and thereafter abbreviate.  For example, spell out *Lithobates* on line 276, but otherwise only use the abbreviation for any *Lithobates* species *unless* it starts a sentence.  Ditto for all other genera.  In many spots where genus is currently spelled out fully, you'll need to abbreviate.  Don't change how you currently have nomenclature in the abstract, any of the tables, or the figure 8 caption.    Line 479-480 (Clean copy).  Delete 'the' before each species (re-abbreviate genus as necessary).  

We look forward to receiving your revised manuscript.

Kind regards,

Dr. Janice L. Bossart

Academic Editor

PLOS ONE
---

## [Author Response · Author response to Decision Letter 3]

21 Aug 2021

On August 2, 2021 I received the following notice.

+++++++++++

PONE-D-20-21124R4

The Louisiana Amphibian Monitoring Program from 1997 to 2017: Results, Analyses, and Lessons Learned.

Dr. Jacoby Carter

Dear Dr. Carter,

We've checked your submission and before we can proceed, we need you to address the following issues:

1.We note that the grant information you provided in the ‘Funding Information’ and ‘Financial Disclosure’ sections do not match.

+++++++++++++++

I think I know where the confusion lies, but I don't know how to address it.

Under 'Financial Disclosure' we indicated we received not specific funding for this work. Which is true.

Under another section I wrote that the project was 'Base Funded'. There is no specific funding for this project. Base funding means our work on this project was considered part of our regular jobs and was covered by our basic salary. There was no specific grant number or project number tied to this work. I am funded through the Ecosystem Missions Area in the USGS. But I don't have a specific grant number I can associate the project with. 

I don't see where I can clarify this in your system.

+++++++++++++++++

To: Janice L. Bossart, Academic Editor

Date: Thursday 29, 2021

Re: Reconciliation Letter for PONE-D-20-21124R1. 

The Louisiana Amphibian Monitoring Program from 1997 to 2017: Results, Analyses, and Lessons Learned.

Dear Dr. Bossart, 

You wrote:

“Dear Dr. Carter,

Thank you for submitting your manuscript to PLOS ONE. After careful consideration, we feel that it has merit but does not fully meet PLOS ONE’s publication criteria as it currently stands. Therefore, we invite you to submit a revised version of the manuscript that addresses the points raised during the review process.

I'm sorry I need to send the manuscript back to you again, but PLoS One manuscripts are not copy edited. Once I accept it will go to the production staff. Just a few things:

• Line 118 (Clean copy), i.e. Table 1 footnote. Although you indicated 'was' had been changed to 'were', it hasn't. Please change, '...data from that route were...'

• Table 1. Please ensure the entire Table can be see in Draft view in Word since it extends past the right margin. I can't doublecheck as I only receive a pdf.

• Scientific nomenclature. I appreciate your effort, but genus is generally only spelled out on first use in the main text and at the start of a sentence and abbreviated otherwise. Sorry, I should have caught this issue earlier, but it only hit me when even more were spelled out in this version. In the main text, spell out genus once and thereafter abbreviate. For example, spell out Lithobates on line 276, but otherwise only use the abbreviation for any Lithobates species unless it starts a sentence. Ditto for all other genera. In many spots where genus is currently spelled out fully, you'll need to abbreviate. Don't change how you currently have nomenclature in the abstract, any of the tables, or the figure 8 caption. 

• Line 479-480 (Clean copy). Delete 'the' before each species (re-abbreviate genus as necessary). 

RESPONSE

I have made the following changes.

1 & 2 I corrected the footnote on line 118. I checked and in Draft View the entire table is visible. Thanks for asking me to check.

3. Okay, I get it now. First use, genus and species. Second use Initial of genus and species, unless the Genus is the first word in the sentence. I went through the manuscript using search feature for the first instance of the full genus-species name and then afterwards to conform. As an additional change, to be consistent, in the few cases where I used the common name after its first introduction I switched to the Latin binomial. 

4. I deleted ‘the’ before the species names in Lines 479-480. The ‘the’ was left over when I changed from ‘…the CommonNameOfSpecies (Latin binomial)…’ to, ‘….L. binomial….’ I reviewed the document and found one other instance where that happened and fixed it.

---

## [Editor Report · Decision Letter 4]

14 Sep 2021

The Louisiana Amphibian Monitoring Program from 1997 to 2017: Results, Analyses, and Lessons Learned.

PONE-D-20-21124R4

Dear Dr. Carter,

We’re pleased to inform you that your manuscript has been judged scientifically suitable for publication and will be formally accepted for publication once it meets all outstanding technical requirements.  Note below under Additional Editor Comments that there are still some minor editorial changes that are required concerning genus and species names.

Kind regards,

Dr.  Janice L. Bossart

Academic Editor

PLOS ONE

Additional Editor Comments:

It seems my directions on how genus and species names should be handled were unclear. Therefore I have copy & pasted example exact text here. To reiterate from my previous decision letter: "For example, spell out *Lithobates* on line 276, but otherwise only use the abbreviation for **any**
*Lithobates* species unless it starts a sentence. Ditto for all other genera."

Hence, lines 294-298 should be: Species could be grouped by when they were heard calling as either winter callers, spring callers, or summer callers (Table 4). Cajun Chorus Frogs (*Pseudacris fouquettei*), Spring Peepers (*P. crucifer*), *L. sphenocephalus*, Crawfish Frogs (*L. areolatus*) and Pickerel Frogs (*L. palustris*) were winter (e.g., Run 1) callers.

Lines 294-298 should NOT be: Species could be grouped by when they were heard calling as either winter callers, spring callers, or summer callers (Table 4). Cajun Chorus Frogs (*Pseudacris fouquettei*), Spring Peepers (*Pseudacris crucifer*), *L. sphenocephalus*, Crawfish Frogs (*Lithobates areolatus*) and Pickerel Frogs (*Lithobates palustris*) were winter (e.g., Run 1) callers.

Please carefully go through the manuscript text and make sure that the genus name is only spelled out the first time it is used and if it begins a sentence. Otherwise, abbreviate the genus name even if it's associated with a different species in the same genus as I've illustrated for lines 294-298 with respect to what the text should vs. should not look like.  Notice genus is abbreviated for all *Lithobates* species listed because it was previously spelled out in earlier text, and is abbreviated for *P. crucifer *because its first use was associated with *Psudacris fouquettei*.  As a reminder, do not change how names are handled in the Abstract, Tables, or Figure caption.
---

## [Editor Report · Acceptance letter]

22 Sep 2021

PONE-D-20-21124R4 

The Louisiana Amphibian Monitoring Program from 1997 to 2017: results, analyses, and lessons learned. 

Dear Dr. Carter:

I'm pleased to inform you that your manuscript has been deemed suitable for publication in PLOS ONE. Congratulations! Your manuscript is now with our production department. 

Kind regards, 

on behalf of

Dr. Janice L. Bossart 

Academic Editor

PLOS ONE